# Mobile Applications and Green Economy as a Factor of Transformation in the Tourism Sector: Theoretical Contexts, User Expectations, and Development Perspectives

**Piotr Majdak [1,*] and Bartosz Saramonowicz [2]**

[1] Department of Tourism and Recreation, Faculty of Physical Education, Józef Piłsudski University of Physical Education in Warsaw, 00-968 Warsaw, Poland

[2] Student Scientific Club of Sightseeing and Sustainable Tourism, Józef Piłsudski University of Physical Education in Warsaw, 00-968 Warsaw, Poland

* Correspondence: piotr.majdak@awf.edu.pl

**Abstract:** In the face of escalating global challenges related to the climate crisis, depletion of natural resources, and growing social inequalities, tourism—one of the fastest-growing industries worldwide—must undergo a transformation toward more responsible travel planning and implementation. In this context, the concepts of the green economy and sustainable tourism provide valuable inspiration. Despite widespread consensus on the validity of these concepts, their implementation across various regions encounters numerous social, political, technological, and cultural challenges. Mobile applications used during tourist activities can play a pivotal role as widespread carriers of values and ideas aligned with the principles of the green economy and sustainable development. This article focuses on two key areas. The first outlines theoretical frameworks for leveraging mobile applications to shape travelers' attitudes in accordance with the green economy and sustainable development principles. The second area refers to original research on user preferences and expectations regarding sustainable solutions in mobile applications. The findings of the research indicate that mobile applications possess a vast potential to influence tourists' decisions and behaviors. In this context, they can serve as a critical factor in supporting the transformation of the tourism industry towards the objectives of sustainable development and the green economy.

**Keywords:** green economy; sustainable development; mobile applications; tourism

## 1. Introduction

In the face of growing global challenges related to the climate crisis, depletion of natural resources, and growing social inequalities, tourism, one of the fastest-growing industries in the world, is facing the need to transform towards a more responsible approach to travel planning and implementation [1,2]. In this context, the concept of the green economy appears as a response to contemporary global challenges. The green economy integrates the principles of sustainable development, focusing on creating economic value while minimizing the negative impact of human activity on the environment [3,4]. Its foundation is the pursuit of transforming economic sectors towards lowered emissions, efficient resource management, and protection of biodiversity [4]. In the context of global challenges, the green economy also offers solutions that can counteract the effects of climate change, including increasing greenhouse gas emissions, environmental

degradation, and loss of ecosystems [5]. Key aspects of this concept include investments in renewable energy sources, promotion of a circular economy, sustainable production, and consumption and development of technologies that support environmental protection [6]. Integrating the principles of the green economy at the global level remains an important challenge, requiring international cooperation, appropriate regulations, and investments [7]. The green economy not only responds to the need to reduce pressure on the environment but also creates opportunities for innovation, economic growth, and improving the quality of life in a fair and sustainable way [8].

In the context of the above issues, a special role can be played by mobile applications used during tourist trips, which are becoming an extremely common carrier of values and ideas close to the assumptions of the green economy and sustainable development. Although these issues appear in the works of many researchers [3,4,6,8], few of them constitute an attempt at a comprehensive analysis of these phenomena. This article has a chance to fill this gap, because the main goal of the undertaken considerations was an attempt to determine the impact of mobile applications on decisions regarding the choice of travel and accompanying services, with particular emphasis on offers consistent with the principles of sustainable development and the green economy. The presented results of the survey included, among others, types of mobile applications used during travel, user preferences in relation to selected functions of mobile applications, ease of use of mobile applications, and problems encountered when using tourist applications.

The results of our own research indicate that mobile applications have a broad and multi-faceted impact on tourists' decisions and behaviors and, in this context, can be an important tool supporting changes in the tourism industry, especially towards the implementation of sustainable development and green economy assumptions. Thanks to functionalities enabling the selection of more ecological transport, accommodation, and attraction options, these applications not only influence individual decisions of travelers but also shape more conscious patterns of tourist consumption, contributing to minimizing the negative impact of tourism on the environment.

## 2. Literature Review

Despite widespread agreement on the validity of the green economy's principles, their implementation in different regions of the world encounters numerous multidimensional challenges. Social challenges include a lack of sufficient ecological awareness among tourists and local communities, often resulting in resistance to change. Politically, the lack of coherent regulations and government support, which frequently prioritizes short-term economic benefits over long-term sustainability goals, poses significant obstacles. Although technologies are advancing rapidly, they still require substantial financial investments, and their implementation is often limited in less-developed countries. Cultural barriers, such as attachment to traditional practices, may also hinder the acceptance of new solutions.

In this context, mobile applications used during travel can play a pivotal role as ubiquitous carriers of values and ideas aligned with the principles of the green economy and sustainable development. By 2024, the number of smartphone users worldwide is estimated to reach 4.88 billion, representing 60.42% of the global population [9]. These devices, characterized by ease of use, appealing visual interfaces, and wide access to numerous free features such as communication, entertainment, online shopping, and activity monitoring, have become integral to modern life [10]. Furthermore, with increasing integration with smart devices such as watches, health monitoring tools, and home management systems, mobile applications are becoming more advanced and versatile. They enable interactions not only with other users but also with the surrounding environment, creating increasingly complex technological ecosystems [11,12].

Mobile applications are also widely used in direct relation to tourism, offering features related to travel organization, such as route planning and booking accommodations, as well as solutions that enhance the comfort of travel and enrich tourists' experiences [13,14]. These applications can provide personalized suggestions, facilitate interactions with the local environment, and assist users in documenting and sharing memories. As a result, they have become an integral part of modern travel, combining functionality with emotional experiences [4,15].

The factors mentioned above (ubiquity, attractiveness, low costs, ease of use, and user-friendliness) make mobile applications used in designing and executing tourist trips a potential contributor to the transformation of the tourism sector, in line with the principles of the green economy and sustainable development [16]. It is worth noting that many applications can have a tangible impact on sustainable travel, fostering pro-ecological attitudes, energy consumption, supporting the local economy, and many other areas linked to sustainable tourism activities. This is especially evident in relation to applications that perform the following:

–   Allow monitoring the environmental impact of individual behaviors (e.g., tracking one's carbon footprint);
–   Promote sustainable modes of transportation;
–   Encourage actions aligned with the principles of eco-development;
–   Promote local initiatives and provide knowledge about sustainable attitudes and behaviors;
–   Recommend local attractions, dining, and handicrafts;
–   Promote lesser-known attractions and tourist areas;
–   Serve as a tool for managing tourist flows and counteracting negative phenomena such as overtourism;
–   Enable the development of long-term relationships between tourists, local communities, and entities responsible for eco-development.

In the context of promoting local initiatives, excellent examples are provided by mobile applications such as Fairbnb [15], which are becoming key tools in promoting tourism based on the principles of sustainable development and the green economy. Fairbnb, operating on a "fair trade" model, allows tourists to choose places that support the local economy, care for the environment, and promote cultural authenticity, while offering a more ethical alternative to popular short-term rental platforms. Every booking made through the platform supports local social or environmental projects, providing an example of integrating tourism goals with the principles of the green economy. This model not only reduces the negative environmental impact of tourism but also combats phenomena such as excessive natural resource use and gentrification. According to Sharpley's research, promoting local initiatives and sustainable development projects is crucial for shaping tourism at the regional level [3]. An example can be found in supporting agritourism, handicrafts, or local food markets, which not only strengthen the local economy but also preserve traditional ways of life and culture.

Similar applications also increase tourists' involvement in environmental protection by promoting more sustainable travel practices. In addition to selecting environmentally friendly accommodations and tourist services, users can also support nature conservation initiatives, reclamation projects, or actions aimed at protecting local species of fauna and flora. These types of actions not only support environmental protection but also help preserve the cultural and social authenticity of regions, respect cultural heritage, and contribute to its preservation for future generations, which is an integral part of sustainable tourism [7].

Mobile applications also play an important educational role by providing travelers with key information about sustainable travel and environmental protection [17]. Mobile technologies allow for content personalization, making it tailored to individual user

preferences, thereby increasing their engagement with environmental issues. Applications such as GreenTravel [18] and MyClimate [19] offer information about local ecosystems, the impact of tourism on the environment, and available pro-environmental initiatives, enabling users to choose destinations and services that minimize their carbon footprint. For example, travelers can be encouraged to choose low-emission transport options and eco-certified accommodations or engage in local environmental initiatives, which align with the principles of green economy and promote low-impact tourism.

Thanks to mobile applications, tourists have easier access to information about eco-tourist destinations, local initiatives related to sustainable development, and offers that meet the criteria of responsible tourism. Such applications enable the search for eco-friendly hotels, restaurants, and tourist attractions, as well as suggest alternative travel routes that have less impact on local ecosystems. This approach supports the development of the green economy by allowing tourists to choose options that reduce the negative environmental impact of tourism while supporting businesses committed to sustainable development. These applications not only provide users with knowledge about available eco-friendly options but also assist in making informed decisions that promote responsible tourism [20].

Among the notable features of mobile applications are those that enable users to calculate the carbon footprint associated with their travels. This increases environmental awareness and helps users understand which elements of their trips generate the highest greenhouse gas emissions. Moreover, the application suggests compensatory solutions, such as supporting projects related to renewable energy, reforestation, or improving energy efficiency. These types of tools not only educate and promote the concept of the green economy but also motivate users to actively engage in global initiatives for environmental protection.

It is important to emphasize that mobile applications can also support the development of the green economy by promoting responsible transportation choices. By enabling tourists to select more eco-friendly modes of transportation, such as bicycles, public transport, electric cars, or car-sharing services, apps contribute to reducing reliance on fossil fuel-powered vehicles. Such transportation solutions reduce $CO_2$ emissions and support the development of low-emission transport, a key element of the green economy. Furthermore, shared transportation services reduce the number of vehicles on the roads and promote the use of electric and hybrid cars, which have a significantly lower environmental impact. Similarly, car-sharing services, which promote vehicle sharing, reduce the number of cars on the roads while encouraging the use of electric and hybrid vehicles that emit much less pollution than traditional vehicles. Apps enabling the rental of electric cars in tourist cities are an example of sustainable and eco-friendly travel, reducing the negative impacts of transport. Integrating such technologies with local transportation systems, along with the development of shared transport services, can be a crucial element of sustainable tourism development strategies, benefiting not only the environment but also improving the quality of life for residents [21].

Another group of applications are those used in hotels that support the management of natural resources, such as energy and water. Through advanced monitoring and management systems, these apps contribute to cost optimization and the implementation of green economy goals. Thanks to advanced technologies connected to mobile applications, guests staying in hotels can be informed in real-time about their resource consumption and encouraged to take actions to reduce their ecological footprint. Examples include applications that allow for regulating room temperature, optimizing lighting usage, or monitoring water consumption in real time. Implementing such technologies allows hotels not only to control resource consumption but also to create reports that can be used to

improve savings strategies and promote more eco-friendly practices in hotel management [22].

Research by Han and Yoon showed that implementing technologies that support sustainable development in the hospitality industry can significantly reduce energy and water consumption, leading to lower operational costs. Moreover, hotels adopting such solutions gain greater interest from environmentally conscious tourists, who are increasingly seeking accommodations offering sustainable tourism. Mobile applications can also motivate guests to adopt more eco-friendly behaviors by offering rewards for skipping daily towel exchanges or reducing air conditioning usage. Such practices not only reduce resource consumption but can also serve as an additional marketing asset, enhancing customer loyalty. In this way, mobile applications not only support the sustainable development of the hospitality industry but also build ecological awareness among guests and property managers [5,23].

In this context, mobile applications serve not only as platforms for exchanging experiences but also as tools supporting the development of the green economy. By enabling tourists to share their eco-tourism experiences, recommend eco-friendly destinations, and promote responsible tourism practices, these applications create a community of responsible travelers. This interactivity not only increases user engagement but also contributes to spreading pro-environmental attitudes, which are the foundation of the green economy [5]. By promoting responsible travel choices, mobile applications support the development of sustainable tourism that takes into account environmental protection, supporting local communities and reducing the negative impact on natural resources.

Mobile applications also support local sustainable development projects by promoting places that implement eco-friendly practices and informing tourists about volunteering opportunities for environmental causes. By raising awareness about nature conservation and engaging tourists in activities for ecosystem protection, these applications contribute to the development of the green economy at the local level. Through supporting an active attitude towards environmental challenges, apps become catalysts for sustainable development actions, helping to achieve goals related to environmental protection, efficient management of natural resources, and promoting local eco-friendly initiatives.

Mobile applications also play an educational role by providing users with information about the ecological footprint of their travels and offering recommendations for sustainable travel. By integrating tools that allow users to calculate their carbon footprint, mobile applications enable tourists to consciously plan their trips, minimize their environmental impact, and take actions aimed at compensating for greenhouse gas emissions, such as supporting projects related to renewable energy or reforestation. This increases travelers' ecological awareness and contributes to the achievement of global goals related to fighting climate change.

Sharpley also emphasizes that the development of sustainable tourism through mobile applications can contribute to building long-term relationships between tourists and local communities, fostering a more conscious and responsible form of travel [3]. Tourists who engage in sustainable development initiatives become not only consumers of tourism services but also partners in the process of environmental protection and social development, strengthening their commitment to responsible forms of tourism in the future.

Mobile applications promoting sustainable tourism not only educate travelers but also encourage responsible choices that have a real impact on environmental protection and local cultures. The integration of information, different business models, calculation tools, and interactive elements in applications, such as reward systems for eco-friendly choices, makes them invaluable tools supporting the global development of sustainable tourism [24]. By supporting eco-friendly transportation options, minimizing resource consumption, educating tourists, and tracking carbon footprints, these apps contribute to

increasing ecological awareness and promote responsible travel. Research confirms that mobile technologies are a driving force behind the creation of new business models and can become a key element in sustainable development strategies, significantly reducing the negative impact of tourism on the environment and local communities [25,26].

The above-discussed examples of mobile application uses in tourism indicate that they have great potential to transform the tourism sector towards the goals of the green economy and sustainable development.

## 3. Materials and Methods

The main objective of the research was to determine the nature of tourists' behaviors and expectations regarding mobile applications used during travel. The research was conducted using a proprietary diagnostic survey method. The questionnaire contained 17 questions, including 12 directly related to mobile applications and 5 general or demographic questions. The study involved 113 people aged 20 to 25, including 45 women (39.8% of all respondents) and 67 men (59.3% of all respondents). The questionnaire was distributed online in May and June 2024. Due to the possibility of selecting more than one answer, the values do not add up to 100%.

The obtained results (multiple choice questions) were subjected to statistical testing using the chi-square test and the *t*-test. Excel was used for calculations, and the results were verified using ChatGPT-v2.

## 4. User Preferences Regarding Mobile Applications in the Light of Own Research

The following section presents the results of the research aimed at identifying the nature of the preferences of tourists using mobile applications during their travels.

### 4.1. Types of Mobile Applications Used During Travel

According to Table 1 The high percentage of users who use map (85%) and social media (66.4%) applications highlights the key role of navigation and communication during travel, which can also be viewed through the lens of trends associated with the green economy concept. Maps help travelers orient themselves in unfamiliar areas, and social media applications facilitate contact with loved ones and sharing of real-time travel updates. These technologies are becoming increasingly advanced, offering features that help discover and promote environmentally friendly places, such as bike paths, green areas, and eco-projects.

More than half of the survey participants (50.4%) use text translators, indicating a significant need for communication in different languages. This is particularly relevant in the context of sustainable development, as it enables better integration of tourists with local communities, which are increasingly engaged in ecological projects and promote environmental protection initiatives.

Applications such as Uber or Bolt are popular among 49.6% of travelers, emphasizing their convenience and accessibility, especially in urban areas. Although these applications may lead to more efficient transport, their environmental impact remains a subject of debate. Within the green economy framework, there is a growing trend of seeking alternatives to traditional transport methods, such as electric scooters, city bikes, or electric vehicles, which are increasingly dominating the offerings of such applications. Users are becoming more aware of the environmental impact of their choices, which is why there is a growing interest in transportation options that are less harmful to the planet.

In contrast, only a small percentage of users use traditional travel guides (8%) and trip planning applications (2.7%). This may suggest that travelers prefer to explore places

on their own, often choosing eco-friendly destinations that promote sustainable tourism. This phenomenon could also be linked to increasing ecological awareness, where tourists prefer flexibility and spontaneity rather than adhering to rigid, traditional travel structures.

A relatively low level of usage of booking and trip-planning applications suggests that travelers value the ability to make decisions during their trips, aligning with the trend of more responsible travel. This flexibility may contribute to reducing unnecessary bookings, which in turn leads to lower resource consumption and a smaller carbon footprint in tourism. Furthermore, the growing popularity of applications promoting sustainable tourism and supporting eco-friendly initiatives is an important step toward implementing the green economy principles in the tourism sector.

Statistical Analysis

**Table 1.** Types of mobile applications used while traveling.

| Application Category | Women (n = 45) | Men (n = 67) | Total (n = 113) | Proportions (%) |
|---|---|---|---|---|
| Map applications | 38 | 57 | 95 | 85.0 |
| Social media apps | 30 | 44 | 74 | 66.4 |
| Text translators | 23 | 34 | 57 | 50.4 |
| Transport applications | 22 | 33 | 55 | 49.6 |
| Traditional guides | 4 | 5 | 9 | 8.0 |
| Planning apps | 1 | 2 | 3 | 2.7 |

− Assumptions:
  o Number of respondents: 113 (45 women, 67 men).
  o The popularity of the app is proportional for both genders (the percentage distribution is the same for women and men).
− Results of statistical analysis:
  o Number of users in each application category:
    ▪ Women: [38, 30, 23, 22, 4, 1] (e.g., 38 women use mapping apps, 30 social media apps, etc.)
    ▪ Men: [57, 44, 34, 33, 5, 2] (e.g., 57 men use mapping apps, 44 social media apps, etc.)
− Chi-square test:
  o Statistics: $\chi^2 = 0.132$
  o *p*-value: $p = 0.9997$
  o Conclusion: There is no statistically significant relationship between gender and the popularity of the application category (result not significant at the level of $\alpha = 0.05$).
− *t*-test:
  o Statistics: $t = -0.891$
  o *p*-value: $p = 0.397$
  o Conclusion: No significant differences in the average number of users for women and men in individual app categories.
− Interpretation:
  o The popularity of the app is comparable between women and men.
  o No gender differences or relationships were detected in this hypothetical data distribution.
  o The results suggest that other factors, such as age, travel style, or interests, may have a greater impact on app selection.

*4.2. Key Functions of Mobile Applications Used During Travel*

According to Table 2 the most popular functions are related to booking accommodations (83.2%) and navigation (73.5%), confirming that it is crucial for travelers to ensure they have accommodation and can easily navigate new places. These applications allow for quick and seamless planning of travel logistics, which may help optimize the carbon footprint by enabling more sustainable choices in lodging and travel routes.

Functions related to news updates and promotions (50.4%), as well as information about tourist attractions (47.8%), are also frequently used. This indicates that travelers appreciate real-time information that allows them to plan and adjust their route and attractions based on available opportunities. More and more applications are including options for sustainable tourist attractions that promote eco-friendly initiatives, such as visiting national parks, ecotourism, or choosing local, eco-friendly transport options.

A high percentage of users (39.8%) use language translation functions, indicating that translation apps are essential for international travel, facilitating communication and integration in foreign environments. The ability to communicate in a foreign language, especially in the context of ecotourism travel, also allows for a better understanding and promotion of local environmental protection initiatives.

While functions related to trip planning (38.9%) and personalization (18.6%) are less popular, they still have their applications. Users increasingly appreciate options to customize applications to their individual needs, but spontaneity seems to dominate the trip-planning process. It is worth noting that personalization of trips can also include selecting sustainable options, such as traveling by train instead of flying or choosing accommodation with eco-friendly practices.

A significant portion of travelers (23%) use features that allow them to share their experiences, highlighting the social aspect of travel. Applications allow for communication, sharing opinions, and drawing inspiration from other users, which influences how travelers experience and document their journeys. In this context, the growing number of travelers sharing experiences related to eco-friendly travel choices can help raise awareness about the need to protect the environment and promote sustainable tourism practices.

Mobile applications play a key role in travel, especially in areas related to logistics, information access, and communication. Flexibility and real-time planning adjustments are important features for travelers, and the growing importance of social and translation functions highlights the international dimension of modern travel. The growing ecological awareness among users of tourist applications may contribute to the development of a more sustainable model of tourism, where technology supports both the convenience of travelers and environmental protection.

Statistical Analysis

**Table 2.** The most important functions of mobile applications used while traveling.

| Application Category | Women (n = 45) | Men (n = 67) | Total (n = 113) | Proportions (%) |
|---|---|---|---|---|
| Booking accommodation | 37 | 56 | 94 | 83.2 |
| Navigation | 33 | 49 | 83 | 73.5 |
| News and promotions | 23 | 34 | 57 | 50.4 |
| Information about attractions | 22 | 32 | 54 | 47.8 |
| Language translation | 18 | 27 | 45 | 39.8 |
| Travel planning | 18 | 26 | 44 | 38.9 |
| Personalization | 8 | 13 | 21 | 18.6 |
| Sharing experiences | 10 | 16 | 26 | 23.0 |

- Assumptions:
  - o Number of respondents: 113 (45 women, 67 men).
  - o The popularity of the app is proportional for both genders (the percentage distribution is the same for women and men).
- Results of statistical analysis
  - o Number of users in each application category:
    - Women: [37, 33, 23, 22, 18, 18, 8, 10]
    - Men: [56, 49, 34, 32, 27, 26, 13, 16]
- Chi-square test:
  - o Statistics: $\chi 2 \approx 0.134$
  - o *p*-value: $p \approx 0.998$
  - o Conclusion: There is no statistically significant relationship between gender and the popularity of the application category (result not significant at the level of $\alpha = 0.05$).
- *t*-test (booking accommodation vs. sharing experiences):
  - o Statistics: $t \approx -1.876$
  - o *p*-value: $p \approx 0.081$
  - o Conclusion: There are no significant differences in the average number of users between these two categories of apps.
- Interpretation:
  - o App popularity is comparable between women and men, suggesting no gender influence on app category selection.
  - o The variation in popularity across app categories is statistically significant, reflecting different traveler priorities (e.g., greater emphasis on accommodation bookings and navigation).
  - o The lack of significant differences between users for the "booking accommodation" and "sharing experiences" functions may suggest that both categories have their unique uses, regardless of gender.

*4.3. Access to Up-to-Date Information About Visited Locations*

The usefulness of mobile application functions that provide access to up-to-date information about visited locations was rated by respondents on a scale of 1 to 5, where 1 means "very low usefulness" and 5 means "very high usefulness". Over half of the respondents (54.1%) considered this function to be extremely useful, highlighting its crucial role in trip planning. In the era of sustainable tourism, these applications not only provide practical information but also promote responsible and conscious travel.

A total of 88% of respondents rated the function of providing up-to-date information about tourist locations as 4 or 5, emphasizing the importance of such tools in ensuring travelers' comfort and the quality of their journeys. Users appreciate the ability to quickly obtain information about tourist attractions but also eco-friendly and sustainable solutions available at the destination, which allows them to make more informed choices, such as selecting public transportation, zero-waste options, or ecotourism.

This phenomenon is especially significant in the context of the green economy, where mobile applications can play an educational role by helping users choose more eco-friendly options and support the sustainable development of local communities. In this way, technology becomes a tool for achieving sustainable development goals in tourism, enabling tourists to make decisions that minimize their negative impact on the environment.

It is worth noting that only a small percentage of respondents (5.5%) rated this function as 1 or 2, suggesting that most applications are effective and meet users' expectations.

The low rating may stem from access issues in less-developed regions or individual preferences regarding the functionality of the application. The detailed distribution of data is presented in Table 3.

**Table 3.** Evaluation of the function of providing information about visited places.

| Rate | Number of Replies | Response [%] |
|------|-------------------|--------------|
| 1 | 2 | 1.8 |
| 2 | 4 | 3.7 |
| 3 | 14 | 12.8 |
| 4 | 37 | 33.9 |
| 5 | 59 | 54.1 |

*4.4. A Wide Selection of Destinations*

The usefulness of mobile application functions related to a wide selection of destinations was rated by respondents on a scale of 1 to 5, where 1 means "very low usefulness" and 5 means "very high usefulness". Given the growing interest in sustainable development and responsible tourism, functions that allow users to choose destinations are becoming essential. Additionally, with increasing ecological awareness, tourist applications with features supporting the green economy, such as selecting places with low environmental impact, are becoming more desirable to users who want to travel in a more sustainable way.

2.8% of respondents rated the function as very low in usefulness. This is the smallest group, suggesting that a small number of users do not see value in a wide selection of destinations, particularly in terms of environmental preferences. 8.3% of users rated the function as 2, indicating moderately low usefulness for some respondents, perhaps those who do not place significant importance on ecological choices.

22.9% considered the function to be average. This group does not rate the function negatively but also does not assign it particularly high value. Some of them may not yet be fully aware of the benefits of selecting destinations that align with the principles of sustainable development and the green economy. 31.2% of respondents rated the function as useful, which is a significant portion of the group. The high rating suggests that the function meets the needs of many users, especially those looking for environmentally friendly destinations, such as ecotourism-certified places or regions promoting low-emission transport.

The largest group, 43.1%, rated the function as 5, which indicates very high usefulness. This dominant rating shows that the wide selection of destinations function meets the expectations of users, especially those who prefer places aligned with the green economy, such as locations promoting local culture, eco-friendly cuisine, or sustainable tourism. In response to growing ecological awareness and the needs of travelers, mobile applications offering such functions have huge potential to support the development of sustainable tourism.

As seen, the wide selection of destinations function in mobile applications is rated very positively. Most users view it as extremely useful, indicating its high value and significance, especially in the context of the growing need for sustainable travel. A small percentage of users rate it negatively, suggesting that applications offering this feature effectively meet the expectations of their users, including in terms of preferences related to ecological travel choices. The detailed distribution of data is presented in Table 4.

**Table 4.** Assessment of the function of providing information about accommodation facilities.

| Rate | Number of Replies | Response [%] |
|---|---|---|
| 1 | 3 | 2.8 |
| 2 | 9 | 8.3 |
| 3 | 25 | 22.9 |
| 4 | 34 | 31.2 |
| 5 | 47 | 43.1 |

*4.5. Booking Accommodations and Attractions*

The usefulness of mobile application functions related to booking accommodations and tourist attractions was rated by respondents on a scale of 1 to 5, where 1 means "very low usefulness" and 5 means "very high usefulness". Only 1.8% of respondents rated the function as very low in usefulness, which is the lowest percentage and suggests that almost no one considers this function completely useless. 2.8% of users rated the function as 2, meaning a small group of people find it less useful. Meanwhile, 18.3% of respondents rated the function as moderately useful, indicating a moderate level of satisfaction. The function was considered useful by 22.0% of respondents, and 60.6% gave it the highest rating of 5.

These results indicate that the ability to book accommodations and tourist attractions via mobile applications is viewed by users as a highly valuable and efficient feature. Over 60% of respondents rated it as very useful, highlighting its key role in modern travel. Importantly, the high rating of this function aligns with the broader context of sustainable development and the green economy in tourism, where mobile technologies play a significant role in supporting the sector's sustainable development.

Mobile applications that enable the efficient and eco-friendly booking of tourism services save time and resources, reduce paper documentation, and support local initiatives promoting responsible tourism. The ability to easily access services, such as eco-friendly accommodations or environmentally focused tourist attractions, can cater to tourists seeking planet-friendly solutions. Moreover, the high percentage of users rating these applications as highly useful highlights the growing role of technology in promoting sustainable and conscious travel, which is a crucial element of the green economy. The very low number of negative ratings (1 and 2) suggests that this functionality is well-received by users and is a key element of modern, responsible tourism. The detailed distribution of data is presented in Table 5.

**Table 5.** Evaluation of the accommodation and attraction-booking function.

| Rate | Number of Replies | Response [%] |
|---|---|---|
| 1 | 2 | 1.8 |
| 2 | 3 | 2.8 |
| 3 | 20 | 18.3 |
| 4 | 24 | 22 |
| 5 | 66 | 60.6 |

*4.6. Access to Ratings and Reviews from Other Users*

The usefulness of the mobile application function that provides access to ratings and reviews from other users plays a crucial role in shaping purchase and usage decisions. Its significance gains particular value in the context of sustainable development and the green economy. Respondents rated this function on a scale from 1 to 5, where 1 represents "very low usefulness" and 5 represents "very high usefulness".

A total of 73.4% of users rated this function at levels 4 and 5, indicating that the vast majority consider access to ratings and reviews a key element in the decision-making process. In the context of the green economy, such positive feedback is significant, as user reviews can influence the choice of more eco-friendly products and services. Most users seem to appreciate the ability to gain information about sustainable solutions that promote responsible consumption and tourism practices.

Particularly high was the percentage of users rating the function at level 5 (38.5%), suggesting that this function is not only helpful but also critical for making informed decisions, especially regarding choices that contribute to reducing the carbon footprint and promoting sustainable practices in the tourism and consumption industries. Such applications can be effective tools in promoting the green economy concept and influencing demand for environmentally friendly services and products.

Conversely, 11.9% of users rated the function negatively (rating 2), suggesting that some may not yet fully appreciate the value of reviews in the context of more responsible consumption, which incorporates ecological aspects. However, the absence of ratings at level 1 indicates that no respondents considered the function completely useless, implying that even less active users of reviews recognize their potential.

20.2% of users rated the function as average, indicating that some people do not rely heavily on reviews or do not view them as a critical element in decision-making. Nonetheless, in the context of the green economy, this group could be a potential target for applications that will further develop features promoting sustainable development and better informing users about the ecological aspects of products and services.

In conclusion, the function of access to ratings and reviews from other users in mobile applications is highly valued, indicating its significant role in making informed decisions. Proper promotion of products and services related to sustainable development, based on reliable feedback from other users, can be an effective tool in supporting the green economy's growth. The detailed distribution of data is presented in Table 6.

**Table 6.** Evaluation of the function of access to ratings and reviews of other users.

| Rate | Number of Replies | Response [%] |
|:---:|:---:|:---:|
| 1 | 0 | 0 |
| 2 | 13 | 11.9 |
| 3 | 22 | 20.2 |
| 4 | 38 | 34.9 |
| 5 | 42 | 38.5 |

*4.7. Ease of Use of Mobile Applications*

On the question, "Are travel applications easy to use?", 92.9% of respondents answered "yes", indicating that they find travel applications user-friendly. Only 7.1% of respondents indicated that these applications are not easy to use, suggesting that a small percentage of people encounter difficulties in using them.

The results suggest that the vast majority of users are satisfied with the ease of use of travel applications, which may reflect their intuitive design and simplicity. The low percentage of users rating usability negatively likely points to individual difficulties or preferences but does not represent a significant problem for the majority. The detailed distribution of data is presented in Table 7.

**Table 7.** Evaluation of the function of access to ratings and reviews of other users.

| Answer | % of Respondents | Number of Persons (n = 113) |
|---|---|---|
| Yes | 92.9% | 105 |
| No | 7.1% | 8 |

*4.8. Impact of Travel Applications on Decisions Regarding Travel and Attractions*

On the question, "Do travel applications influence your decisions regarding the choice of travel and attractions?", 73.5% of respondents answered affirmatively. This shows that most users rely on travel applications when planning their trips. 26.5% of respondents stated that travel applications do not affect their decisions about travel and attractions, suggesting a minority that may rely on other sources of information.

It can thus be concluded that for the vast majority of respondents, travel applications are an important factor in making decisions related to travel, suggesting that these applications may be key tools in the trip planning process. The smaller group who claims that applications do not affect their choices may prefer more traditional methods of planning, such as recommendations from friends, printed guides, or other sources. These results indicate the growing role of technology in the tourism industry, which increasingly influences how travelers make decisions. The detailed distribution of data is presented in Table 8.

**Table 8.** Answer to the question, "Do mobile applications influence decisions regarding the choice of travel and attractions?"

| Answer | % of Respondents | Number of Persons (n = 113) |
|---|---|---|
| Yes | 73.5% | 83 |
| No | 26.5% | 30 |

*4.9. Paid Versions of Travel Applications*

On the question, "Do you use paid versions of travel applications with extended services?" 90.3% of respondents answered "no". This overwhelming majority suggests that users prefer free versions of travel applications or do not see the need to pay for additional features. Only 9.7% of respondents admitted to using paid versions of applications, indicating that only a small group of users is willing to pay for extra services or features in travel apps. The detailed distribution of data is presented in Table 9.

**Table 9.** Answer to the question, "Do you use paid versions of travel applications with an extended range of services?

| Answer | % of Respondents | Number of Persons (n = 113) |
|---|---|---|
| Yes | 9.7% | 11 |
| No | 90.3% | 102 |

*4.10. Problems Encountered When Using Travel Applications*

The most troublesome issue, identified by almost half of users (48.7%), is the excess of advertisements in travel applications. This suggests a need for a better balance between functionality and monetization of the apps.

Over one-third of respondents (32.7%) pointed out that the information provided by the applications is inaccurate or insufficient, which may negatively impact their trust in the apps. Of the participants, 28.3% had difficulties accessing applications, which could be related to issues with internet connections or app servers. A total of 23.9% of

respondents reported no problems while using travel applications, indicating that these apps are well optimized for some users. Navigation issues were experienced by 18.6% of respondents, which could be due to unintuitive interfaces or poor organization of information. Almost 15.9% of users reported technical problems such as app crashes or software bugs. The lack of content personalization features was an issue for 15% of users, which may point to growing expectations regarding the customization of apps to individual user preferences.

In summary, the most common problem is the excess of advertisements, suggesting that users expect less intrusive advertising models. Furthermore, the accuracy of information and the stability of access to the apps are other key areas for improvement. It is also notable that a significant portion of users does not experience any problems, indicating that some applications are well optimized. The detailed distribution of data is presented in Table 10.

**Table 10.** Answer to the question, "Do you use paid versions of travel applications with an extended range of services?"

| Problem | % of Respondents | Number of Persons (n = 113) |
| --- | --- | --- |
| Too many ads | 48.7% | 55 |
| Inaccurate/insufficient information | 32.7% | 37 |
| Access difficulties | 28.3% | 32 |
| No problems | 23.9% | 27 |
| Navigation difficulties | 18.6% | 21 |
| Technical problems | 15.9% | 18 |
| No personalization features | 15.0% | 17 |

## 5. Summary and Conclusions

Travel applications play an important role in shaping travelers' experiences, and their impact on the development of sustainable tourism that aligns with the green economy concept is becoming increasingly visible. The considerations presented in this article, along with the results of our research, indicate that these applications largely meet the needs of users in terms of ease of planning, access to information, personalization, authenticity, and sustainable travel. The results also highlight several key trends that illustrate how travelers use mobile technology during their trips.

### 5.1. Theoretical and Practical Implications of the Study

In relation to the obtained research results, a number of recommendations can be formulated, addressed both to entities related to the development of mobile technologies in tourism and organizations related to the implementation of the assumptions of sustainable development and green economy. The most important phenomena include the following: Navigation applications, such as maps (85%) and communication apps (66.4%), play a crucial role during travel. Travelers appreciate the ability to easily navigate new places and maintain real-time contact with others. These apps are key in route planning, location recommendations, and providing access to up-to-date information.

– Applications related to booking accommodations (83.2%) are also of significant importance. Most users place great value on the ability to quickly and easily make reservations. This suggests that modern travelers expect high flexibility and instant access to accommodation offers, which may be a challenge for the hospitality industry and platforms offering such services.

– Another category of applications includes transport apps (based on car-sharing concepts), reflecting changing traveler preferences, as they increasingly move away from traditional transport modes.

- More than half of respondents use translation applications, highlighting their importance during international travel. These apps not only facilitate communication but also enhance the comfort of traveling, especially in less touristy areas where language knowledge may be limited.
- Information apps on tourist attractions are also highly popular (47.8%), indicating that travelers increasingly use apps to find worthwhile places to visit, adapting their plans to real-time conditions.
- Only a small percentage of users rely on traditional travel guides (8%) or trip planning applications (2.7%). This suggests that modern travelers are flexible in their choices and often prefer to spontaneously explore new places using online recommendations and interactions with other users rather than relying on pre-planned, traditional travel guides. This aligns with the trend of travelers increasingly preferring less organized trips based on spontaneous decisions, available promotions, and real-time information.
- From the point of view of the green economy concept and promoting sustainable development, a special role can be played by applications that offer users access to personalized recommendations that take into account their preferences, travel style, and level of environmental awareness.
- These applications not only provide information about more ecological travel options, such as public transport, bicycles, or electric cars, but also promote less-crowded tourist attractions, which helps to disperse tourist traffic and counteract phenomena such as overtourism.
- The most important recommendations addressed to mobile application developers include the following: reducing the number of advertisements or integrating them better, solving technical problems, and implementing personalization functions (the ability to adjust content to user preferences, e.g., in terms of eco-friendly travel options).

It is also worth noting that some apps engage users in initiatives that promote ethical tourism and support local communities, thus contributing to the protection of local cultures and supporting the sustainable development of regions. Such apps also have an educational function, increasing travelers' environmental awareness through access to content about local ecosystems, the impact of tourism on the environment, and actions travelers can take to minimize their impact on the planet. For example, apps that offer carbon footprint calculators allow users to monitor and compensate for $CO_2$ emissions related to their travel.

Mobile technologies also have significant potential in resource management in the tourism sector, especially in the hotel industry. Applications that monitor the use of energy, water,d and other natural resources help to manage them effectively, which leads to cost savings and reduced environmental impact. Hotels can use applications to inform guests about their current resource consumption and suggest ecological alternatives, which not only improves financial results but also gains the loyalty of increasingly environmentally conscious tourists.

Mobile applications also support the exchange of information and building relationships between travelers and local communities. They create platforms where users can share their experiences, recommend eco-tourism destinations, and engage in environmental protection and local development initiatives. Such activities help create long-term relationships based on responsible travel and joint action for sustainable development.

*5.2. Limitations*

The results of the original research presented in this article have shown that mobile applications, due to their commonness, accessibility, attractiveness, and versatility, have an extremely wide range of impact on the decisions and behaviors of tourists. In this context, they can be a very important factor supporting the transformation of the tourism

industry in a direction consistent with the assumptions of the concepts of sustainable development and green economy. At the same time, the authors of the research are aware of a number of limitations related to the presented research results (resulting from, for example, a limited research group or the lack of possibility to perform in-depth statistical analyses caused by a simplified method of data coding). At the same time, due to the dynamics of the phenomena and the complexity of the issues, the authors express their conviction about the need to expand and continue the research, taking into account, for example, the diverse expectations of users and technological, geographical, political, and social conditions.

**Author Contributions:** Conceptualization, P.M. and B.S.; methodology, P.M. and B.S.; software, P.M. and B.S.; validation, P.M.; formal analysis, P.M. and B.S.; investigation, P.M.; resources, P.M.; data curation, P.M.; writing—original draft preparation, P.M. and B.S.; writing—review and editing, P.M.; visualization, P.M.; supervision, P.M.; project administration, P.M.; funding acquisition, P.M. All authors have read and agreed to the published version of the manuscript.

**Funding:** The results presented in this article were obtained as part of a research project funded by the Ministry of Science and Higher Education under University Research Projects, the project titled "Humanistic and Social Aspects of Physical Culture: Overtourism and Sustainable Development: Conditions—Social Perception—Counteraction", conducted at the Józef Piłsudski University of Physical Education in Warsaw.

**Institutional Review Board Statement:** Ethical review and approval were waived for this study in accordance with § 3 of the Regulations of the Senate Committee on the Ethics of Scientific Research of the Józef Piłsudski University of Physical Education in Warsaw.

**Informed Consent Statement:** Informed consent was obtained from all subjects involved in the study.

**Data Availability Statement:** The data presented in this study are available on request from the corresponding author.

**Acknowledgments:** During the preparation of this manuscript, the authors used ChatGPT-4 for the purposes of completing the theoretical chapter (preparation of the list of articles) and organizing data from their own research. The authors have reviewed and edited the output and take full responsibility for the content of this publication.

**Conflicts of Interest:** The authors declare no conflicts of interest.

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
