# Peer review of "Mobile Applications and Green Economy as a Factor of Transformation in the Tourism Sector: Theoretical Contexts, User Expectations, and Development Perspectives"

_sustainability, doi:10.3390/su17031168_

Round 1

Reviewer 1 Report

Comments and Suggestions for Authors

Comments to the Authors:

Dear authors, first I would like to express my gratitude for the opportunity to review your article. After a thorough analysis of the manuscript, I would like to provide some constructive suggestions to enhance the clarity and impact of your work:

General comment: The study is interesting from an academic point of view, but the authors need to significantly improve some points. The authors should present an introduction outlining the gap in the literature that motivated the study. They should also include studies that justify the study and introduce the reader to what is already being done in tourism in terms of innovation. On the other hand, the authors need to detail the methodology they are using, and the methods employed. In addition, the authors need to discuss the results obtained with the literature and present the implications of the study. Finally, they should review the conclusions and present the limitations and future lines of research. After reformulating these points, I believe that the manuscript can improve its scientific contribution and increase its chances of being accepted.

1. Mobile Applications And Green Economy: Towards the Transformation of the Tourism Sector

“Amid escalating global challenges related to the climate crisis, depletion of natural resources, and increasing social inequalities, tourism, one of the fastest-growing indus-tries worldwid, efaces the necessity of transforming towards a more responsible approach to planning and conducting travel. In this context, the concept of the green economy emerges as a response to contemporary global challenges.”

Reviewer: How can they be sure that there is a need to adopt a more sustainable approach? How do they know that the green economy is emerging as a responsibility? These kinds of statements need to be justified with previous literature. Authors should add citations to back up their assertions. On the other hand, “point 1.” of the manuscript is more like a literature review and not an introduction. Although the authors present a broad discussion of the green economy and mobile applications, they do not clearly specify the specific objectives of the study, i.e. what they are actually trying to achieve with the research. Point 1” focuses more on contextualization and the relevance of the topic than on outlining the purpose of the study, i.e. it does not explicitly present the ‘gap’ in knowledge or problem that the study seeks to address. For example, the authors need to refer to studies that point to the need to develop more studies in tourism with a view to including new technologies that help modernize the sector and thereby increase sustainability, such as the study https://doi.org/10.20867/thm.29.2.10 which suggests the importance of new studies on tourism companies that innovate their models using technology. In addition, “point 1” mentions challenges in implementing the green economy and social, political, and cultural barriers, but does not identify a specific point in the literature that is missing or how the study seeks to fill this gap. Although the importance of the topic is inferred by the contextualization of global challenges and the growing relevance of mobile applications, the authors do not explicitly justify why this study is necessary or what the expected impact of the results is. On the other hand, the need for the study is implied by the relevance of the topics covered (sustainability, green economy, and mobile applications in tourism), but the authors do not directly emphasize how the study will contribute to solving practical problems or advancing academic knowledge. Despite mentioning challenges related to the green economy and limitations in adopting sustainable technologies, it is not clear how this study makes a unique contribution to the literature or to practice in the tourism sector. In other words, the authors fail to refer to previous studies that mention what has been done in this area, i.e. what has been done in tourism to make companies' business models more innovative and sustainable? For example, the recent study https://doi.org/10.1016/j.tmp.2024.101235 mentions that tourism companies are seeking to include virtual activities that contribute to the sustainability of companies and the sector itself. In my view, the authors should split up “point 1” and give an introduction, followed by a contextualization or review of the topic. The way in which “point 1” is presented does not clarify the central points of the study and so they should create a section to give more prominence to the key points of the study, referring to studies that help to contextualize the topic and the need for this study.

2. Materials and Methods

Reviewer: The authors need to provide more information on this point. The authors do not clearly specify what type of methodology they used in the study. Also, the age range is limited to young people between 20 and 25, why? On the other hand, it is not mentioned whether the participants were randomly selected or whether they represent a specific population (such as tourists from a region, country or with certain socio-economic characteristics). Furthermore, no information is provided on how the questionnaire was validated (whether it was pre-tested) or on the reliability and validity of the questions used. It is not specified whether the questions are open or closed and whether it has been adapted from the literature. At this point, the authors should also mention the techniques that will be used to analyse the data collected.

3. User preferences regarding mobile applications in the light of own research

Reviewer: I believe that this point is a preliminary assessment of the results based solely on the evaluation of the Likert-type scale. At this point I realize that the authors did not carry out any estimation of the measurement model with structural equations, but only based themselves on percentage values. This issue greatly reduces the scientific relevance of the study. Despite this, the study could be significantly improved if the authors presented a discussion of the results, i.e. compared the results with the existing literature. In this way, the scientific relevance of the study can be seen. In addition, the authors need to present the theoretical and practical implications of the study, since the authors do not present an outstanding scientific component, the authors should present in the implications, guidelines, and possible solutions for tourism companies to facilitate the adoption of technologies and the adoption of more sustainable behaviours. This information should be suggested based on the results obtained.

4. Summary and Conclusions

Reviewer: At this point the authors should only refer to the conclusions of the study. Alternatively, separate the summary from the conclusions. The conclusions presented are exaggerated in relation to the results, i.e. the authors are concluding more than what they obtained in the results. That's why it's important to create the “Implications of the study” section and take these assumptions there. In addition, the authors should add in the conclusions what objectives were initially proposed and what gaps in the literature were answered in this study.  On the other hand, the authors need to identify the limitations of the study and suggest future lines of research.

Author Response

Dear Reviewer,
First, we would like to thank you for your thorough review and all your advice. Below we present detailed responses to all your suggestions.

1. We have extracted and supplemented the chapters containing the introduction and the literature review.

2. We have supplemented the information referring to previous studies and specifying the gap in knowledge.

3. We have supplemented the information on materials and methods (along with detailed information on the method of data collection and the study population).

4. We have supplemented the statistical calculations.

5. We have supplemented the references to literature, including the sources you indicated to us.

6. In the summary part, we have extracted the section "Theoretical and practical implications of the study" and "Limitations", and we have also supplemented information on future research directions.

Thank you again for all your comments and suggestions. We hope that the corrections we have introduced will meet with your understanding and acceptance.

Reviewer 2 Report

Comments and Suggestions for Authors

Dear authors,

1. The section 1 seems to be too long. Preferably divide it intro two, the intorduction and the literature review. At the end of the introduction please write the scope/purpose of your reseach and describe the sections to follow and then continue with your literature review.

2. Section 2: Your sample was 113 people between 20-25. Why did you select this age gap? What is the reason behind that? 113 people from where? And why only 113 people, is there a reason behind it? Your answer, if scientific, should be also added in the limitations of your research.

3. Section 3: Which methodology did you use? I can only see descriptive statistics without any chart or diagram. Which statistical package did you use? There are no correlations between your variables. In a scientific paper we cannot exact conclusions only from the discriptive statistics. You need to indicate whether your measures are reliable and valid for the study. There are many types of tests you need to perform, eg. regression analysis, correlations, chi-square tests, t-tests, anova, multivariate analysis whatever suits best to your research or a combination. 

4. Limitations is a significant part of a research especially with small sample. 

Concluding there is a serious lack of the methodology and this must be improved.

Author Response

Dear Reviewer,
First, we would like to thank you for your thorough review and all your advice. Below we present detailed responses to all your suggestions.

1. We have extracted and supplemented the sections containing the introduction and the literature review.

2. We have supplemented the information on materials and methods (along with detailed information on the method of data collection and the study population).

3. We have subjected the results of our own research (multiple choice questions) to statistical analyses (chi-square test, t-test).

4. In the summary part, we have related the results of the analyses to the research of other authors, extracted the section "theoretical and practical implications of the study", and supplemented the information on the limitations and future directions of research.

Once again, thank you for all your comments and suggestions. We hope that the corrections we have introduced will meet with your understanding and acceptance.

Reviewer 3 Report

Comments and Suggestions for Authors

As mobile applications become an integral part of modern travel and play a pivotal role as ubiquitous carriers of values and ideas aligned with the principles of the green economy and sustainable development, it can serve as a critical factor in supporting the transformation of the tourism industry towards the objectives of sustainable development and the green economy. It is an interesting research topic.

This research was conducted using a proprietary diagnostic survey method to outline theoretical frameworks for leveraging mobile applications to shape travelers’ attitudes in accordance with the green economy and sustainable development principles. At last, it has present the results of the research aimed at identifying the nature of the preferences of tourists using mobile applications during their travels.

Author Response

Dear Reviewer,

Thank you very much for your review.

Best regards –

Piotr Majdak

Round 2

Reviewer 1 Report

Comments and Suggestions for Authors

Dear authors,

The new version of the manuscript has significantly improved the scientific contribution, clarifying the importance of carrying out this study and the need for further research. Given the significant improvement of the article, I believe it can be accepted for publication.

Good luck!

Reviewer 2 Report

Comments and Suggestions for Authors

Nothing else to mention.